# A Multivariate Causal Discovery based on Post-Nonlinear Model

**Kento Uemura**                                        UEMURA.KENTO@FUJITSU.COM
**Takuya Takagi**                                       TAKAGI.TAKUYA@FUJITSU.COM
*Fujitsu Ltd., Japan*
*RIKEN Center for Advanced Intelligence Project, Japan*

**Takayuki Kambayashi**                         KAMBAYASHI.TAKAYUKI@RYOBI.CO.JP
*Ryobi Systems Co., Ltd., Japan*

**Hiroyuki Yoshida**                         HIROYUKI.YOSHIDA@JAPAN-SYSTEMS.CO.JP
*Japan Systems Co.,Ltd, Japan*

**Shohei Shimizu**                      SHOHEI-SHIMIZU@BIWAKO.SHIGA-U.AC.JP
*Shiga University, Japan*
*RIKEN Center for Advanced Intelligence Project, Japan*

**Editors:** Bernhard Schölkopf, Caroline Uhler and Kun Zhang

## Abstract

Understanding causal relations of systems is a fundamental problem in science. The study of causal discovery aims to infer the underlying causal structure from uncontrolled observational samples. One major approach is to assume that causal structures follow structural equation models (SEMs), such as the additive noise model (ANM) and the post-nonlinear (PNL) model, and to identify these causal structures by estimating the SEMs. Although the PNL model is the most general SEM for causal discovery, its estimation method has not been well-developed except for the bivariate case. In this paper, we propose a new causal discovery method based on the multivariate PNL model. We extend the bivariate method to estimate multi-cause PNL models and combine it with the iterative sink search scheme used for the ANM. We apply the proposed method to synthetic and real-world causal discovery problems and show its effectiveness.

**Keywords:** multivariate causal discovery, structural equation models, post-nonlinear causal model

## 1. Introduction

Understanding causal structures of systems is a fundamental question in science and has been studied in various fields such as biology, economics, and social science (Rhein and Strimmer, 2007; Londei et al., 2006; Morgan and Winship, 2014; Moneta et al., 2013). While conducting randomized experiments is the most effective way to identify causal structures, it is often impossible for ethical, technical or cost reasons (Spirtes and Zhang, 2016). Therefore, it is important to develop causal discovery methods that infer causal structures from uncontrolled observational data.

Using structural equation models (SEMs) is one major approach in causal discovery. In SEM-based methods, causal structures are assumed to follow parameterized causal models and inferred by estimating the parameters from data. An important property of causal SEMs is identifiability that is a theoretical guarantee that true causal structures can be uniquely identifiable if the joint distributions of data are known. Constructing flexible identifiable models and then developing their estimation methods are the main focuses in SEM-based study.

Various identifiable SEMs and their estimation methods have been proposed so far. The linear non-Gaussian acyclic model (LiNGAM) (Shimizu et al., 2006) is one of the most studied models. In

LiNGAM, a causal structure is represented as a directed acyclic graph (DAG) and each effect variable is generated as the linear combination of cause variables followed by an additive unobserved noise that is independent of the causes. There has been proposed an efficient estimation method named DirectLiNGAM (Shimizu et al., 2011; Hyvärinen and Smith, 2013). While LiNGAM is a simple and well-studied model, it may suffer from degradation in complex real-world applications due to the linear assumption. The additive noise model (ANM) is proposed as a more flexible model that can handle nonlinear causal relations (Hoyer et al., 2009). In ANM, cause variables are transformed nonlinearly and then added to an unobserved noise independent from the causes. ANM is proved identifiable except for some trivial cases and an estimation method based on regressions, named regression with subsequent independence test (RESIT), is proposed (Hoyer et al., 2009; Peters et al., 2014). As a more realistic model, Zhang and Hyvärinen (2010) proposed the post-nonlinear (PNL) model inspired by real-world data observation processes. In the PNL model, cause variables are transformed and added to a noise variable in the same manner as ANM and then transformed again by a nonlinear invertible function. The second transformation represents the sensor or measurement distortion, which is frequently encountered in practice. Although the PNL model is one of the most flexible identifiable causal SEMs, which covers ANM and LiNGAM, its estimation methods are not well-developed except the bivariate case (Zhang and Hyvärinen, 2009; Zhang et al., 2015; Uemura and Shimizu, 2020; Tu et al., 2021).

Uemura and Shimizu (2020) proposed an estimation method of the bivariate PNL model and a causal discovery procedure based on it, named autoencoder-based causal discovery for the PNL model (AbPNL). AbPNL represents the two nonlinearities with neural networks and estimates them by minimizing two loss terms corresponding to two assumptions of the model: the independence of the cause and the noise and the invertibility of the second nonlinearity. The losses are defined as Hilbert-Schmidt independence criterion (HSIC) (Gretton et al., 2005) and a reconstruction loss, respectively, and minimized directly by a stochastic gradient descent method. Given samples of two variables, AbPNL estimates the models of the two candidate causal directions and infer the true direction as the one that fits the model better. While AbPNL is easy to use owing to no assumptions on neural network structures and the straightforward optimization procedure, it is limited to the bivariate cases and cannot be applied to multivariate cases unless an exhaustive search is performed, which is usually infeasible in practice.

In this paper, we propose a new method for multivariate nonlinear causal discovery based on the PNL model. First, we extend the estimation of the bivariate PNL model of AbPNL to the multi-cause model with multiple cause variables and a single effect variable. Then, we iterate the multi-cause model estimation to identify the causal order of the variables in the same manner as the ANM case in RESIT. Finally, we prune redundant edges of the causal DAG constructed from the causal order to obtain the true causal structure. We also show the theoretical validity of the proposed procedure in the multivariate PNL model. We apply the proposed method to synthetic and real-world datasets and show its effectiveness.

## 2. Preliminaries

### 2.1. Multivariate Causal Discovery

In this paper, we address the following multivariate causal discovery problem. Let $d$ variables $X := \{x_1, x_2, \ldots, x_d\}$ with a density $p(x_1, x_2, \ldots, x_d)$ have causal relations represented as a directed graph with an adjacency matrix $\mathbf{A} := [a_{i,j}]_{d \times d} \in \{0, 1\}^{d \times d}$, where $x_j$ is a direct cause of $x_i$ if and

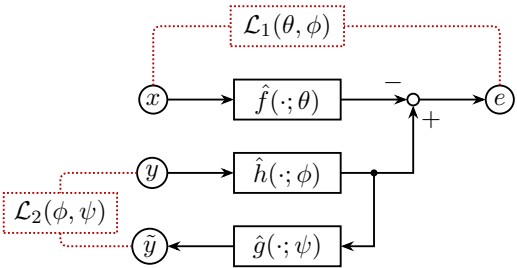

Figure 1: Network structure to estimate the bivariate PNL model in AbPNL. Circles and solid boxes are variables and neural networks, respectively. Dotted boxes represent loss terms with involving variables.

only if $a_{i,j} = 1$. Our goal is to estimate $\mathbf{A}$ from observations $\mathcal{D} := \{(x_1^{(s)}, x_2^{(s)}, \ldots, x_d^{(s)})\}_{s=1,\ldots,n}$ sampled i.i.d. from $p(x_1, x_2, \ldots, x_d)$. We assume that the causal graph is a directed acyclic graph (DAG), *i.e.* there is no cycle of directed edges on the graph.

## 2.2. Post-Nonlinear Causal Model

Zhang and Hyvärinen (2010) proposed the post-nonlinear (PNL) causal model taking account of real-world data generating processes. In the PNL model, variables are generated as follows:

$$x_i = \begin{cases} g_i(f_i(X_{\mathrm{pa}_i}) + e_i) & \text{if } X_{\mathrm{pa}_i} \neq \emptyset, \\ e_i & \text{otherwise,} \end{cases} \quad (i = 1, \ldots d) \tag{1}$$

where $X_{\mathrm{pa}_i}$ is a set of direct cause variables, or parents, of $x_i$ and $e_i$ is an unobserved noise variable. The inner function $f_i : \mathbb{R}^{|X_{\mathrm{pa}_i}|} \to \mathbb{R}$ represents nonlinear mixture of the causes and the outer one $g_i : \mathbb{R} \to \mathbb{R}$ represents nonlinear distortion of sensors that often observed in real-world situations. The noises are independent of each other, $e_i \perp e_j \ (\forall i \neq j)$, and thus, a noise is independent of each of the corresponding causes, $e_i \perp x_j \in X_{\mathrm{pa}_i}$. The distortion $g_i$ is assumed to be invertible. The PNL model is identifiable except for some special combinations of the functions and the noise distributions (Zhang and Hyvärinen, 2009; Peters et al., 2011, 2014). The PNL model is known as the most general identifiable causal model and is equivalent to LiNGAM when $f$ is linear and $g$ is the identity and ANM when $g$ is the identity.

## 2.3. Autoencoder-based causal discovery for PNL model (AbPNL)

Uemura and Shimizu (2020) proposed a bivariate causal discovery method named AbPNL. AbPNL assumes that a causal relation follows the bivariate PNL model. Given two variables $(x_1, x_2)$, AbPNL estimates two models corresponding to two candidate causal directions, $x_1 \to x_2$ and $x_1 \leftarrow x_2$, and infers that the one with the better model fit is the correct direction based on the identifiability of the model.

In the estimation process of a model, AbPNL uses neural networks to represent the functions and trains them by minimizing a loss consisting of two terms corresponding to two assumptions of the model: the independence between the cause and the noise and the invertibility of the distortion. Suppose two variables $\{x, y\}$ have the causal direction $x \to y$, and consider the estimation of the

bivariate PNL model,

$$y = g(f(x) + e). \tag{2}$$

Figure 1 shows the network structure for the estimation. By the invertibility of $g$, the noise in (2) can be written as

$$e = g^{-1}(y) - f(x). \tag{3}$$

AbPNL uses neural networks $\hat{f}$ and $\hat{h}$ for $f$ and $g^{-1}$ in (3), respectively. The first loss term encodes the independence between $x$ and $e$, which is measured by the empirical estimator of Hilbert-Schmidt independence criterion (HSIC) (Gretton et al., 2005). Let $k$ and $l$ be Gaussian kernel functions for $x$ and $e$, respectively, and $\{(x^{(s)}, y^{(s)})\}_{s=1,\dots m}$ be the given samples, the loss is defined as

$$\mathcal{L}_1(\theta, \phi) := \mathrm{HSIC}_b(\{(x^{(s)}, e^{(s)})\}_{s=1,\dots,m}) := \frac{1}{m^2} \, \mathrm{trace}(\mathbf{KHLH}), \tag{4}$$

where $e^{(s)} := \hat{h}(y^{(s)}; \phi) - \hat{f}(x^{(s)}; \theta)$, $\mathbf{K} := [k(x^{(s_1)}, x^{(s_2)})]_{m \times m}$ and $\mathbf{L} := [l(e^{(s_1)}, e^{(s_2)})]_{m \times m}$ are Gram matrices, $\mathbf{H} := \mathbf{I} - m^{-1}\mathbf{1}\mathbf{1}^{\mathrm{T}}$, and $\mathbf{1}$ is an $m$-dimensional vector of ones. HSIC takes a non-negative value and zero if and only if two variables are independent. The second loss term encodes the invertibility of $g$. By representing $g$ with a neural network $\hat{g}$ as well as $g^{-1}$, the second loss term is defined as the reconstruction error of $y$,

$$\mathcal{L}_2(\phi, \psi) := \frac{1}{m} \sum_{s=1}^{m} \left| y^{(s)} - \tilde{y}^{(s)} \right|^2, \tag{5}$$

where $\tilde{y}^{(s)} := \hat{g}(\hat{h}(y^{(s)}; \phi); \psi)$ is a reconstructed sample of $y^{(s)}$. The final loss function is defined as

$$\mathcal{L}(\theta, \phi, \psi) := (1 - \lambda)\mathcal{L}_1(\theta, \phi) + \lambda\mathcal{L}_2(\phi, \psi), \tag{6}$$

where $\lambda \in (0, 1)$ is a balancing weight.

## 3. Proposed Method

In this section, we propose a new multivariate nonlinear causal discovery method based on the PNL model. Our basic idea is to extend the estimation method of the bivariate PNL model in AbPNL to the multi-cause model and estimate the multivariate model by following the principle for the multivariate ANM used in RESIT (Peters et al., 2014).

In the following sections, we first develop the estimation method of the multi-cause PNL model that has multiple causes and a single effect by modifying the bivariate one in AbPNL. Then, we introduce the procedure to identify the causal order of variables by iterating multi-cause estimations and the theoretical validity. Finally, we describe the pruning method to construct a causal graph from the causal order.

### 3.1. Estimation of the multi-cause PNL model

We extend the bivariate estimation procedure in AbPNL described in Section 2.3 to the multi-cause PNL model defined as

$$y = g(f(X) + e), \tag{7}$$

---

**Algorithm 1:** IdentifyCausalOrder

---

**input**  : Samples of $d$ variables $X = \{x_1, \dots, x_d\}$
**output** : Causal order $\pi$
$R \leftarrow \{1, \dots, d\}, \pi \leftarrow []$
**repeat**
    **for** $j \in R$ **do**
        Estimate (7) with causes $\{x_i\}_{i \in R \setminus \{j\}}$ and effect $x_j$.
        $q_j \leftarrow$ degree of model fit
    **end**
    $j^* \leftarrow \operatorname{argmax}_j q_j$
    $\pi \leftarrow [j^*, \pi]$
    $R \leftarrow R \setminus \{j^*\}$
**until** $|R| = 1$
$\pi \leftarrow [R(1), \pi]$

---

where $X$ is a set of causes, $y$ is an effect, and $e$ is an unobserved noise independent of $\forall x \in X$. The difference from the bivariate model in (2) is the number of causes. Therefore, we can use most parts of the bivariate algorithm except the function $f$ and the first loss term $\mathcal{L}_1$. We deal with the former simply by increasing the input dimension of the network $\hat{f}$ to $|X|$. For the latter, the loss should quantify the degree of dependence between the noise and each cause. To achieve this, we define the multi-independence loss term as

$$\mathcal{L}_1(\theta, \phi) := \max_{x \in X} \operatorname{HSIC}_b(\{x^{(s)}, e^{(s)}\}). \tag{8}$$

Although other aggregation schemes such as mean and median could be used, we employ the maximum to make all HSIC values decrease evenly. Note that our modification is the generalization of the original bivariate estimation, which is in the case of $|X| = 1$.

### 3.2. Identification of causal order

To identify the causal order of variables, we follow the iterative procedure proposed by Peters et al. (2014). While the procedure is introduced for the multivariate ANM, it is also valid on the multivariate PNL model by the following theorem.

**Theorem 1** *Suppose variables $X$ follow a multivariate PNL model on DAG $\mathcal{G}$. Then there exists a multi-cause PNL model with an effect $x \in X$ and causes $X \setminus \{x\}$ if and only if $x$ is a sink on $\mathcal{G}$.*

The "if" part is obvious from the definition. The converse can be easily verified in the same manner as the ANM case (see Peters et al., 2014, A. 15)[1]. Based on the theorem, we can find a sink variable on the true causal graph with estimations of multi-cause PNL models.

Algorithm 1 shows the pseudocode for identifying the causal order. Given $d$ variables $X$, we first estimate $d$ possible multi-cause models with $(d-1)$ causes. Then, we select the effect of the model that satisfies the PNL assumptions the best as a sink variable and exclude it from the variable set. By iterating these steps, we identify the causal order.

---

1. For more detailed proof, see Appendix A.

---

**Algorithm 2:** PruneRedundantEdges

---

**input** : Samples of $d$ variables $X = \{x_1, \ldots, x_d\}$ and causal order $\pi$
**output** : Adjacency matrix $\mathbf{A} = \{a_{i,j}\}_{d \times d} \in \{0,1\}^{d \times d}$
$\mathbf{A} \leftarrow \mathbf{O}$
**for** $k \in \{2, \ldots, d\}$ **do**
   $i \leftarrow \pi(k), C \leftarrow \pi(1 : k - 1)$
   **if** $|C| \geq 2$ **then**
      **for** $j' \in C$ **do**
         Estimate (7) with causes $\{x_j\}_{j \in C \setminus \{j'\}}$ and effect $x_i$.
         **if** *the model satisfy the assumptions* **then**
            $C \leftarrow C \setminus \{j'\}$                     /* Remove the edge $x_{j'} \to x_i$. */
         **end**
      **end**
   **else**
      **if** $x_i$ *is independent of* $x_{C(1)}$ **then**
         $C \leftarrow \emptyset$                         /* Remove the edge $x_{C(1)} \to x_i$. */
      **end**
   **end**
   **for** $j \in C$ **do**
      $a_{i,j} \leftarrow 1$
   **end**
**end**

---

We use residual independence loss values $\mathcal{L}_1$ for evaluating the goodness of model fit. To reduce the effect of the overfitting, we calculate the values from test samples that are not used for the model estimation. Although the reconstruction loss $\mathcal{L}_2$ should also be considered for fair evaluation, it showed a small enough value in almost all the preliminary experiments. Therefore, we pay attention to the independence loss, which is more difficult to minimize[2].

### 3.3. Pruning of redundant edges

To remove redundant edges on a DAG constructed from the identified causal order, we apply the greedy pruning procedure (Peters et al., 2014). Algorithm 2 shows the pseudocode. The overall flow is the same except that we modify the method of determining to prune to suit the PNL model.

Considering the causal DAG in which each variable is a direct cause of variables with later causal order, we greedily remove unnecessary edges on each of its multi-cause substructures. The greedy pruning procedure is done as follows. On a multi-cause substructure, we search for a cause $x_{j'}$ such that the other causes $\{x_j\}_{j \in C \setminus \{j'\}}$ and the effect $x_i$ still follow the PNL model if the cause is removed. If such a cause exists, we remove the corresponding edge and repeat the procedure until all the causes are checked.

---

2. Nevertheless, there is no theoretical guarantee that $\mathcal{L}_2$ always becomes small enough. Thus, in the following experiments, we estimate one model multiple times and take the median over results with small enough values of $\mathcal{L}_2$. It also takes effect on stabilizing results of stochastic optimization.

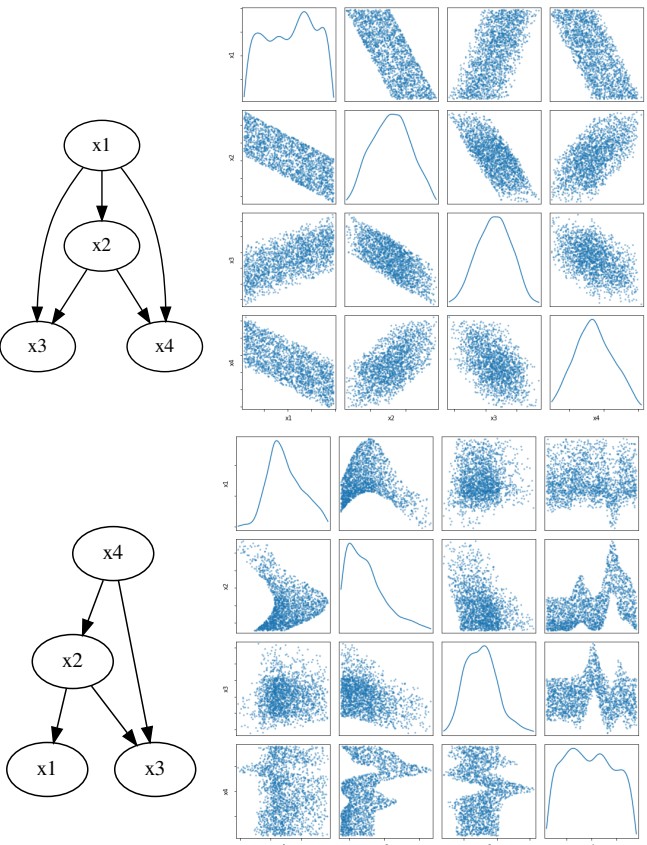

Figure 2: Examples of randomly generated DAGs and samples (upper: linear, lower: nonlinear).

When there are one or more causes, we evaluate the goodness of model fit by checking whether the independence assumption holds, as in Section 3.2. However, in contrast to the relative evaluation in the sink identification, we use absolute criteria. It is because if a model $y = g(f(x)+e)$ holds, the model with another variable $y = g'(f'(x, z) + e')$ also holds. It may cause the failure in removing $z$ in practice if we use the relative measure such as a difference of loss values. Therefore, we use the independence test of HSIC (Gretton et al., 2008) between the estimated noise and the excluded cause. When there is no cause, we use the test between the effect and the excluded cause.

## 4. Experiments

In this section, we conduct numerical experiments and show the effectiveness of the proposed method on synthetic and real-world multivariate causal discovery problems.

### 4.1. Synthetic data

To investigate the empirical performance of the proposed method, we generated synthetic problems with the ground truths and evaluated the performance of the proposed method. We randomly generated causal structures with linear and nonlinear causal relations. For each problem, we first

| | Linear Problem | | Nonlinear Problem | |
|---|---|---|---|---|
| | $d_{\mathrm{edit}}$ | $N_{\mathrm{rev}}$ | $d_{\mathrm{edit}}$ | $N_{\mathrm{rev}}$ |
| Proposed | $0.81 \pm 1.20$ | $0.19 \pm 0.51$ | $\mathbf{2.16 \pm 1.55}$ | $\mathbf{0.59 \pm 0.70}$ |
| RESIT(Linear) | $0.23 \pm 0.47$ | $0.01 \pm 0.10$ | $3.80 \pm 1.91$ | $1.11 \pm 0.96$ |
| RESIT(GAM) | $0.24 \pm 0.47$ | $0.01 \pm 0.10$ | $3.21 \pm 1.90$ | $0.78 \pm 0.87$ |
| RESIT(GP) | $0.26 \pm 0.52$ | $0.01 \pm 0.10$ | $3.22 \pm 1.87$ | $0.77 \pm 0.86$ |
| DirectLiNGAM | $\mathbf{0.21 \pm 0.43}$ | $\mathbf{0.00 \pm 0.00}$ | $4.44 \pm 2.64$ | $1.52 \pm 1.37$ |

Table 1: Results of synthetic problems. Numbers are the averages and the standard deviations of the edit distance $d_{\mathrm{edit}}$ and the number of reversed edges $N_{\mathrm{rev}}$ over 100 synthetic problems.

constructed a causal DAG with $d$ nodes by generating each edge with the probability[3] of $2/(d-1)$, and allocated the variables for the nodes randomly. We used $d = 4$ in this experiment. Then, for a linear problem, we assigned linear causal relations, $x_i = \sum_{x_j \in X_{\mathrm{pa}_i}} \beta_{ij} x_j + e_i$ with uniformly chosen coefficients $\beta_{ij} \sim U(-1, 1)$. For a nonlinear problem, we generated the PNL model in (1), where $f_i$ and $g_i$ were sampled as the weighted sums of Gaussian processes and sigmoid functions, respectively. We adjusted scales of variables so that signal-to-noise ratios become one to avoid extreme cases. We generated 100 linear and 100 nonlinear causal structures and, on each structure, 2000 samples from uniform noises with the zero mean and the unit deviation. Figure 2 shows examples of linear and nonlinear problems with their ground truth DAGs.

We followed the settings used in Uemura and Shimizu (2020) including the hyperparameters and structures of neural networks. We used a half of samples for the model estimation and the rest for the evaluation of model fit. In the identification of a sink, we estimated each model for $t = 9$ times and used the median values of $\mathcal{L}_2$ of test samples over trials that achieve $\mathcal{L}_1 < 10^{-3}$. Similarly, in the pruning, we remove an edge if $\mathcal{L}_1 < 10^{-3}$ achieved and the independence test ($p = 5\%$) passed in at least one estimation out of $t$. As compared methods, we used DirectLiNGAM (Shimizu et al., 2011) and RESIT (Peters et al., 2014) with linear, GAM and GP regressions.

Table 1 shows the results. As performance measures, we used the edit distance and the number of reversed edges,

$$d_{\mathrm{edit}}(\mathbf{A}, \mathbf{B}) := \sum_{i,j} |a_{i,j} - b_{i,j}|, \tag{9}$$

$$N_{\mathrm{rev}}(\mathbf{A}, \mathbf{B}) := \sum_{i,j} a_{i,j} \times b_{j,i}, \tag{10}$$

where $\mathbf{A} := [a_{i,j}]_{d \times d} \in \{0, 1\}^{d \times d}$ and $\mathbf{B} := [b_{i,j}]_{d \times d} \in \{0, 1\}^{d \times d}$ are an estimated adjacency matrix and its ground truth, respectively. While $d_{\mathrm{edit}}$ represents the performance of the overall estimation, $N_{\mathrm{rev}}$ measures that of the causal order identification more directly.

On the linear problems, all methods show good performance with $d_{\mathrm{edit}} < 1$. Intuitively, DirectLiNGAM achieved the best and the performance decreases as the assumed models become more complex. Although the proposed method took the largest edit distance, the difference from the best is 0.6, which we think is still practically good enough. From the value of $N_{\mathrm{rev}}$, we can find that RESIT and DirectLiNGAM estimated correct causal orders and their misestimation is mainly

---

3. Under the probability, the expected number of edges on a DAG is $d$.

|  | sim1 | | sim2 | |
| --- | --- | --- | --- | --- |
|  | $d_{\mathrm{edit}}$ | $N_{\mathrm{rev}}$ | $d_{\mathrm{edit}}$ | $N_{\mathrm{rev}}$ |
| Proposed | 0 | 0 | 3 | 1 |
| RESIT(GAM) | 7 | 1 | 15 | 2 |
| DirectLiNGAM | 3 | 0 | 6 | 0 |

Table 2: Results of fMRI simulation data. Numbers are the edit distance $d_{\mathrm{edit}}$ and the number of reversed edges $N_{\mathrm{rev}}$ on sim1 and sim2 datasets.

due to the pruning process. On the other hand, the proposed method failed in estimating the causal order on some problems than other methods.

On the nonlinear problems, the result shows the opposite trend. DirectLiNGAM and RESIT with the linear regression deteriorated because of the violation of their linear assumption. By virtue of the nonlinearity, RESITs with GAM and GP regression improved the performance. The proposed method improved the performance of DirectLiNGAM by 2.28 and RESIT by 1.05 in terms of the edit distance. Considering that the expected number of edges in each problem is $d = 4$, we conclude that the improvement is significant.

### 4.2. fMRI simulation data

To evaluate the proposed method in a more realistic scenario, we applied it to simulated fMRI data (Smith et al., 2011). We used sim1 and sim2 datasets with 5 and 10 variables, respectively. Each dataset consists of 50 time series and each series has 200 data points, which results in 10000 data points in total. We used randomly chosen 5000 points and used 1000 as test samples in the proposed method.

Table 2 and Figure 3 show the results. For RESIT, we only show the result of the GAM regression that outperformed the others. While all the three methods are comparable in terms of $N_{\mathrm{rev}}$, the proposed method showed the best performance in $d_{\mathrm{edit}}$. These facts and the estimated DAGs in the figure suggest that the proposed method works correctly not only in the causal order identification but also in the pruning procedure.

### 4.3. General Social Survey data

Finally, we applied the proposed method to General Social Survey data used in Shimizu et al. (2011) and analyzed the result on a real-world problem. This sociological data consists of 1380 samples and each sample has 6 variables: Father's education, Father's occupation, Number of siblings, Son's education, Son's occupation and Son's income. It is challenging to estimate their causal relations because they may violate the model assumption such as the independence of noises. To obtain results with higher reliability, we estimated a DAG 10 times with bootstrapping (Efron and Tibshirani, 1993) and output the edges that appeared in at least half of all estimations. We set the number of bootstrap samples to 600.

Figure 4 shows the estimated DAG. Although the true causal structure is unknowable, the proposed method estimated the direction from father to son, which is the only undeniable true direction. Moreover, the direction from son's occupation to son's income seems to be reasonable. On the other hand, there are directions contrary to our intuition such as the one from occupation to education. In

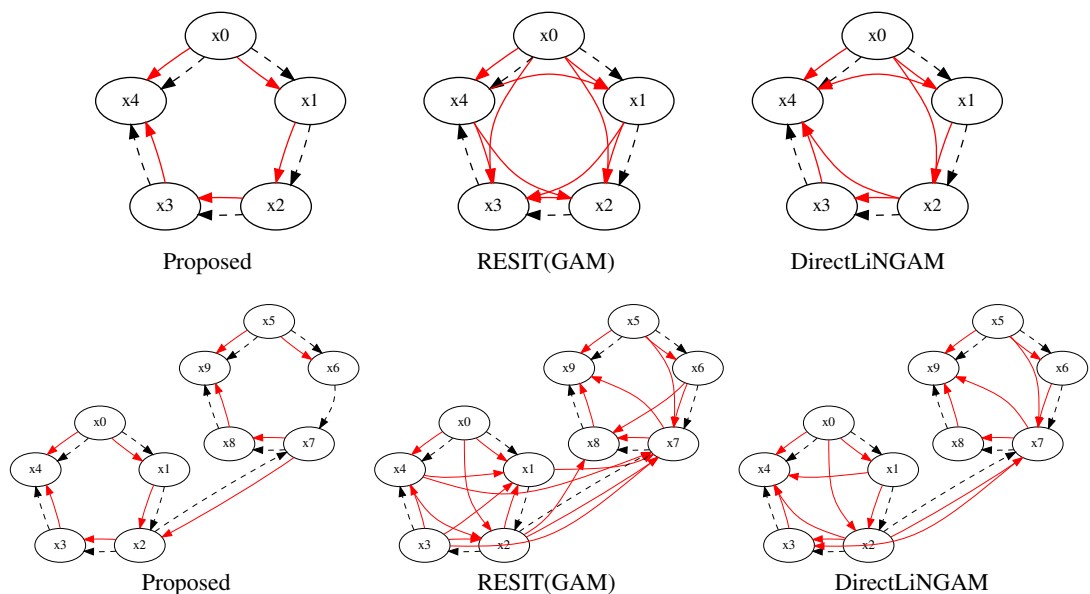

Figure 3: Estimated DAGs from fMRI simulation data (upper: sim1, lower: sim2). Solid and dashed lines are estimated edges and ground truths, respectively.

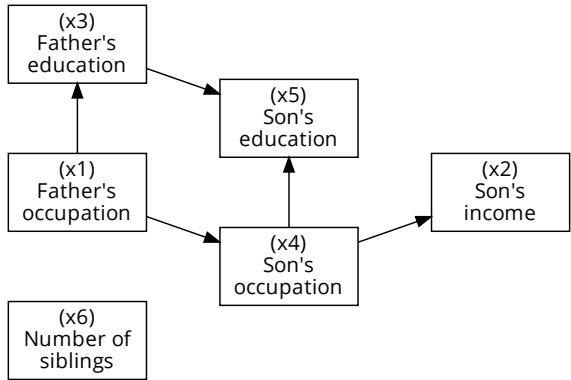

Figure 4: Estimated DAG from GSS data.

the case where the model assumptions are violated, a more detailed analysis of the proposed method is necessary to ensure robustness, which we think is important future work.

## 5. Discussion

In this section, we discuss the properties of the proposed method, specifically the computational complexity and the stabilization.

In the proposed method, we use neural networks to represent models and estimate them by optimizing their parameters using stochastic gradient descent methods. Since each neural network represents a simple scalar function, it is unnecessary to be a complex design. Therefore, the train-

ing time is much shorter than the general deep models such as image recognition and language processing.

On the other hand, the number of estimations grows exponentially as the number of variables increases. Given $n$ variables, the identification of causal order requires $n - 1$ iterations. In each iteration, we compare $k$ models for the remaining $k$ variables. Therefore, a total of $n + (n - 1) + \cdots + 2 = n(n - 1)/2 - 1$ model estimations are required, which is the same as RESIT and DirectLiNGAM. Additionally, the proposed method trains one model several times to stabilize results. Consequently, the total number of training becomes $t(n(n-1)/2-1)$, where $t$ is the number of training trials per one model. Similar to the identification of causal order, the pruning procedure also requires an exponential number of estimations. For the worst case, where all the edges will be pruned, a total of $t(k(k - 1)/2 - 1)$ estimations are required for each multi-cause substructure with $k$ cause variables[4]. The number of estimations increases as the ground truth DAG becomes sparse.

One effective approach to alleviate the computational time is parallelization. Since all the estimations in one iteration of the identification of causal order can be performed separately, we can reduce the time by parallelizing them. If enough computational resources are available, we can identify the causal order with $n - 1$ serial iterations. Similarly, the time of the pruning can be reduced by the same manner. In the experiment in Section 4, we used this parallelization. We additionally applied the proposed method to synthetic problems with $n = 15$ and confirmed that it requires roughly one day for one problem on a server capable of (maximum) 40-threading[5].

Estimating one model multiple times in the proposed method has a high impact on the total computational time. Therefore, we think improving the stability is an important issue that should be addressed not only for improving the performance of causal discovery but also for reducing the computational time. One of the typical approaches is to change the optimizing algorithm, while there is a trade-off between stability and speed. Another possible option is to add a regularization term to the loss. To use this idea, we need to consider carefully the impact of changing the loss on using its value when comparing models in the sink identification.

## 6. Conclusion

In this paper, we proposed a new multivariate nonlinear causal discovery method based on the PNL model. We extended the bivariate estimation method to the multi-cause one and combined it with the iterative schema for identifying the causal order. We then applied the greedy pruning method to construct a resulting DAG. We showed the effectiveness of the proposed method with synthetic and simulated fMRI data and analyzed the result on the real-world sociological data. We discussed the computational cost of the proposed method and suggested parallelization as one alleviation. A more fundamental improvement of stability is an important issue that needs to be addressed in the future. As suggested in Section 4.3, a comprehensive analysis of the impact of model violation is also necessary to ensure its performance on real-world applications.

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

|  | Linear Problem | | Nonlinear Problem | |
|---|---|---|---|---|
|  | $d_{\text{edit}}$ | $N_{\text{rev}}$ | $d_{\text{edit}}$ | $N_{\text{rev}}$ |
| Proposed | $11.72 \pm 6.89$ | $2.55 \pm 1.98$ | $20.04 \pm 5.83$ | $4.43 \pm 2.00$ |
| RESIT(Linear) | $4.00 \pm 3.00$ | $0.08 \pm 0.27$ | $35.45 \pm 11.12$ | $5.40 \pm 2.05$ |
| RESIT(GAM) | $3.96 \pm 2.91$ | $0.08 \pm 0.27$ | $26.08 \pm 8.04$ | $3.65 \pm 1.74$ |
| RESIT(GP) | $67.24 \pm 6.88$ | $1.92 \pm 1.29$ | $28.19 \pm 8.93$ | $3.82 \pm 1.78$ |
| DirectLiNGAM | $2.41 \pm 1.62$ | $0.00 \pm 0.00$ | $28.75 \pm 8.43$ | $6.15 \pm 2.58$ |

Table 3: Additional results of synthetic problems ($n = 15$). Numbers are the averages and the standard deviations of the edit distance $d_{\text{edit}}$ and the number of reversed edges $N_{\text{rev}}$ over 100 synthetic problems.

R. Tu, K. Zhang, H. Kjellstrom, and C. Zhang. Optimal transport for causal discovery. In *ICML 2021 workshop on the Neglected Assumptions in Causal Inference*, July 2021.

K. Uemura and S. Shimizu. Estimation of post-nonlinear causal models using autoencoding structure. In *ICASSP 2020 - 2020 IEEE International Conference on Acoustics, Speech and Signal Processing (ICASSP)*, pages 3312–3316, 2020.

K. Zhang and A. Hyvärinen. On the identifiability of the post-nonlinear causal model. In *Proc. 25th Conference on Uncertainty in Artificial Intelligence (UAI2009)*, pages 647–655, 2009.

K. Zhang and A. Hyvärinen. Distinguishing causes from effects using nonlinear acyclic causal models. In *Proc. Workshop on Causality: Objectives and Assessment at NIPS 2008*, pages 157–164, 2010.

K. Zhang, Z. Wang, J. Zhang, and B. Schölkopf. On estimation of functional causal models: General results and application to the post-nonlinear causal model. *ACM Trans. Intell. Syst. Technol.*, 7 (2), 2015.

## Appendix A. Proof of Theorem 1

In the following, we show detailed sketch of the proof of Theorem 1. For more strict proof, see (Peters et al., 2014).

For "if" part, let $x$ be a sink variable and $X'$ be the other variables. From the definition of the multivariate PNL model (1), there exists a PNL model such that $x = g(f(X') + e)$, where $e \perp x' \in X'$. Note that non-parent variables of $x$ are ignored in terms of the function $f$.

For "only if" part, suppose there exists a PNL model with an effect $y$ that is not a sink on the true DAG $\mathcal{G}$. Since $y$ is not a sink, $y$ has children on $\mathcal{G}$. Let $z$ be a sink on the subgraph consisting of all children of $y$ and $D$ be descendants of $z$. Since $D$ does not have children of $y$, $D \perp y \mid S \cup \{z\}$ holds, where $S = X \setminus \{y, z\} \cup D$. Therefore, from the assumption, there exists a PNL model $y = g'(f'(S, z) + e')$. On the other hand, since $z$ is a child of $y$, there exists a PNL model $z = g(f(S, y) + e)$. Given the situation in which $S$ was observed, this contradicts the identifiability of the bivariate PNL model.

## Appendix B. Additional experiments on 15 variables

To investigate the applicability of the proposed method on a larger problem, we applied it to synthetic problems used in Section 4.1 with $n = 15$. We used the same parameter settings of both the problem generation and the proposed method, which are designed and tuned based on smaller problems. We conducted the experiments on Intel Xeon E5-2690 v4 2.60GHz servers with a maximum of 40 parallelizations. The application to one problem was finished in about one day. Table 3 shows the average results. While DirectLiNGAM outperforms the others on linear problems, the proposed method shows better performance on nonlinear problems. Note that since the parameters of the proposed method were tuned on smaller problems, we need additional experiments and analysis for a more comprehensive evaluation.

