# OpenReview forum: "A Multivariate Causal Discovery based on Post-Nonlinear Model"
_cclear.cc/CLeaR/2022/Conference — CLeaR 2022 Poster_

### Official Review · Reviewer_wyQs · 2021-11-18

**Confidence:** 5
**Overall Score:** 6

**Main Review:**

The paper is clearly written and organized. But I have the following concerns:
1. Novelty: the paper has some small innovation points, but can not give us valuable enlightenment. It is more a combination of some existing methods/tricks with tiny adaption.
2. Theoritical foundation: the paper lacks a systematic proof of model identifiability and inference consistency.
3. Experiments: For the experiments on both simulation and real data, the datasets are too simple with just several variables, how the performace will be if running on large datasets? The propsed method relies on learning many DNN models to evaluate data fitting, however, when the variable and sample scale are large, the time efficiency and hyperparameter adjusting will be a problem. The author claimed the proposed method is a general one which can process both linear and nonlinear relations, however, results in table I do not support such cliam. For the accuracy showed in table 2, how much is from the deep models used in the new method, and how much is from the causal order learning?

**Summary:**

This paper extended the AbPNL method developed for bivariate causal discovery to process multivariated case. It first borrowed a widely used  causal order learning method, and combined it with multivariate AbPNL to learn a causal order. An edge pruning method is then used to get a DAG based on the causal order.

---

> ### Author Response · Authors · 2021-12-04
> **Response to Reviewer wyQs (1/2)**
>
> Thank you for reviewing our paper and the insightful comments.
>
> We would like to answer your questions and commens in the following. We will reflect the answers in our camera-ready version.
>
> > Novelty: the paper has some small innovation points, but can not give us valuable enlightenment. It is more a combination of some existing methods/tricks with tiny adaption.
>
> In this paper, we aim to develop a new multivariate causal discovery method based on the PNL model. Despite its theoretical foundations, the practical estimation method for the multivariate PNL model is not well developed. We believe that our proposal increases the applicability of SEM-based causal discovery methods and promotes their use in real-world problems.
>
> > Theoritical foundation: the paper lacks a systematic proof of model identifiability and inference consistency.
>
> The identifiability of the multivariate PNL model is already shown in the existing studies as stated in Section 2.2. We focused on the development of its estimation method in this paper.
>
> For the inference consistency of RESIT, Theorem 34 in [Peters 14a] says, "RESIT used with a consistent non-parametric regression method and an independence oracle is guaranteed to find the correct graph from the joint distribution." Since our method uses the iterative sink identification and the greedy pruning schema used in RESIT, the correct graph can be estimated if the consistency of the autoencoding part, i.e. the invertibility of the outer function, is guaranteed.
>
> > Experiments: For the experiments on both simulation and real data, the datasets are too simple with just several variables, how the performace will be if running on large datasets? The propsed method relies on learning many DNN models to evaluate data fitting, however, when the variable and sample scale are large, the time efficiency and hyperparameter adjusting will be a problem.
>
> Thank you for the important comment on the scalability of our method. As you pointed out, the scale of the problem such as the number of variables has an impact on the performance. The iterative sink identification requires n(n-1)/2-1 model estimations, where n is the number of variables, and thus the computational time grows when the number of variables is large. Additionally, in our method, we conducted the training of neural networks several times for one model to stabilize the results, which further increases the time. Since all the trainings in one iteration can be performed separately, we used parallel computation to reduce the time in the experiments.
>
> For the time of a single training, the neural networks in our method represent simple scalar functions and do not need to have complex structures. Therefore, the training time is much shorter than the general deep models such as image recognition and language processing. We think that hyperparameters such as network structures and the selection of optimizers are insensitive for the same reason and we used the same parameters used in [Uemura and Shimizu 20] throughout all the experiments.
>
> We will add the Discussion section to the paper and put these discussions. Furthermore, we conducted additional experiments on larger problems of n=15. The following results show the performance of 10 generated problems due to the time limitation. We will increase the number of problems to evaluate the performance in more general and put the results in the camera-ready version with additional analyzes of the results.
>
> |               |  d_edit (L) |   N_rev (L) |  d_edit (N) |   N_rev (N) |
> | ------------- | -----------:| -----------:| -----------:| -----------:|
> | AbPNL(0.95)   | 11.9 (5.43) | 2.40 (1.71) | 19.2 (4.49) | 3.50 (1.27) |
> | RESIT(linear) | 2.80 (2.04) | 0.00 (0.00) | 39.2 (12.4) | 5.20 (1.93) |
> | RESIT(gam)    |  3.3 (1.57) | 0.00 (0.00) | 29.9 (9.17) | 3.90 (2.23) |
> | RESIT(gp)     | 68.0 (8.01) | 1.60 (1.17) | 32.5 (7.69) | 4.10 (1.52) |
> | DirectLiNGAM  |  2.5 (1.78) | 0.00 (0.00) | 31.8 (9.54) | 6.70 (2.71) |
>
> (L): Results on 10 linear problems, (N): Results on 10 nonlinear problems.

---

> > ### Author Response · Authors · 2021-12-04
> > **Response to Reviewer wyQs (2/2)**
> >
> > > The author claimed the proposed method is a general one which can process both linear and nonlinear relations, however, results in table I do not support such cliam.
> >
> > As you pointed out, the results in Table 1 show that our method is superior to the other methods on nonlinear problems and inferior on linear problems. However, we believe that causal relations of complex real-world systems are not fit in such a simple linear model and the results on nonlinear problems are more important to evaluate the performance in practice.
> >
> >
> > > For the accuracy showed in table 2, how much is from the deep models used in the new method, and how much is from the causal order learning?
> >
> > The difference between the results of our method and RESIT suggests the effectiveness of using the deep models because they adopt the same causal order learning scheme. Although the difference between the values of d_edit is large, the difference is mainly caused by the performance of the pruning part as inferred from Fig. 3. On the other hand, the values of N_rev show the performance of causal order learning more directly. The difference suggests that our method could approximate the causal relations more precisely and thus, estimate the true causal order by adopting deep models.

---

### Official Review · Reviewer_k2Ax · 2021-11-19

**Confidence:** 4
**Overall Score:** 8

**Main Review:**

The paper considers an essential problem of estimating causal structure from observation data. \
The paper builds on prior work in the field of post-nonlinear models. The authors put some effort into estimating the whole graph when the data follow the PNL model.\
The critical challenge is how to estimate PNL efficiently. The authors use neural networks to represent and optimize the two functions of PNL. \
The theoretical analysis or experimental results are presented in a logical way.

The way of literately finding a sink is my only concern. This way will affect the performance of your method when the number of variables is large. It would be nice to discuss this issue in the main paper.

**Summary:**

This paper considers the problem of learning causal structure from observed data generated by the post-nonlinear model. The authors first study the estimation of the PNL model with multi-parents by autoencoder strategy of Neural network. Then, the authors propose a two-step method to estimate the post-nonlinear model, including identifying causal order and removing redundant edges.

---

> ### Author Response · Authors · 2021-12-04
> **Response to Reviewer k2Ax**
>
> Thank you for reviewing our paper.
>
> We would like to answer your qustion. We will reflect the reponse in our camera-ready version.
>
> > The way of literately finding a sink is my only concern. This way will affect the performance of your method when the number of variables is large. It would be nice to discuss this issue in the main paper.
>
> Thank you for your suggestion. We will set up the Discussion section and add more detailed discussions.
>
> The computational cost in the iterative sink identification grows rapidly as the number of variables increases. To be more precise, since the algorithm finds one sink in one iteration, it requires n-1, where n is the number of variables, iterations to identify the causal order. In a single iteration, we compare k models for given k variables.
> Therefore, a total of n+(n-1)+...+2=n(n-1)/2-1 model estimations are required, which is the same as RESIT and DirectLiNGAM. Additionally, in our method, we conducted the training of neural networks several times for one model to stabilize the results. Consequently, the total number of training is t*(n(n-1)/2-1), where t is the number of training trials per one model.
>
> One effective approach to reduce the computational time is parallelization. Since all the trainings in one iteration can be performed separately, we can reduce the time by parallelizing them, which we used in the experiments. For the time of a single training, the neural networks in our method are used to represent simple scalar functions and do not need to have complex structures. Therefore, the training time is much shorter than the general deep models such as image recognition and language processing.
>
> We conducted additional experiments on larger problems of n=15. The following results show the performance of 10 generated problems due to the time limitation. We will increase the number of problems to evaluate the performance in more general and put the results in the camera-ready version with additional analyzes of the results.
>
> |               |  d_edit (L) |   N_rev (L) |  d_edit (N) |   N_rev (N) |
> | ------------- | -----------:| -----------:| -----------:| -----------:|
> | AbPNL(0.95)   | 11.9 (5.43) | 2.40 (1.71) | 19.2 (4.49) | 3.50 (1.27) |
> | RESIT(linear) | 2.80 (2.04) | 0.00 (0.00) | 39.2 (12.4) | 5.20 (1.93) |
> | RESIT(gam)    |  3.3 (1.57) | 0.00 (0.00) | 29.9 (9.17) | 3.90 (2.23) |
> | RESIT(gp)     | 68.0 (8.01) | 1.60 (1.17) | 32.5 (7.69) | 4.10 (1.52) |
> | DirectLiNGAM  |  2.5 (1.78) | 0.00 (0.00) | 31.8 (9.54) | 6.70 (2.71) |
>
> (L): Results on 10 linear problems, (N): Results on 10 nonlinear problems.

---

> > ### Comment · Reviewer_k2Ax · 2021-12-10
> > **Thank you for your response**
> >
> > Thank you for addressing my concerns with this response. I will keep my overall score.

---

### Official Review · Reviewer_rX9F · 2021-11-20

**Confidence:** 3
**Overall Score:** 6

**Main Review:**

The paper follows a straightforward approach: The authors extend the autoencoder-based estimation of the bivariate PNL model by Uemura and Shimizu (2020) to the multi-cause case and then apply a causal order and pruning procedure as proposed for the ANM case in Peters et al. (2014). To learn y = g(f(x) + e) they use an auto-encoder with loss terms for the independence between x and e, here max_i(HSIC(xi, e)) and the reconstruction error of y. The approach is applied to a single (!) DAG model and real-world datasets. Overall a weak accept.


# Pros:

* (Potentially) the first estimator for the multi-cause PNL
* Several steps to aid the stability of estimation are undertaken

# Cons:

* Numerical experiments extremely limited (owing to extreme computational demands)
* Few limitations / weaknesses discussed
* Omitted proofs despite Supplementary Material
* No code provided

# Originality

The authors state that even the publication of the PNL model was already in 2012 there hasn't been work on multi-cause estimators. I found another work ("Post-nonlinear Causal Model with Deep Neural Networks" by Chung etal, 2019) that also addresses a multi-cause PNL with neural networks. Hence, the authors would have to explain the theoretical difference and/or compare them numerically.

# Significance

The PNL model is the to-date most general among the class of restricted structural causal models and may in many cases be the only one applicable given complex datasets. A novel estimator is, hence, an important contribution.

# Quality

The introduction and preliminaries sufficiently cover the problem setting. The proposed method straightforwardly follows from the bivariate previous work and the causal order and pruning approach from the ANM. These are sufficiently described. For the only Theorem the authors state "The “if“ part is obvious from the definition. The converse can be easily verified in the same manner as the ANM case (see Peters et al., 2014, A. 15).". Since there are no space problems here, I would provide the proof for completeness. There are also no guarantees discussed for how well the autoencoder may fit in a general setting.

The authors mention the issue of stabilization in the Conclusions, but could they elaborate on this also in the Discussion?

My main concern is about the numerical comparison. In fact only one single DAG with a few SCMs is considered and the proposed approach compared to the RESIT estimator of ANM and DirectLinGAM. As mentioned above, there seems to be another relevant neural network estimator that could be considered here.

The main issue seems to be the extreme computational complexity of the method. Could the authors discuss how this can be alleviated? Also, except for computational complexity no weaknesses are discussed.

I would also suggest to provide code somewhere.


# Clarity

The style of writing suffers a bit from English grammar issues.


# Further comments

* "Causal discovery methods assume that causal structures follow structural equation models"
What about constraint/score-based methods. These do not assume one particular SEM, rather a very general class. I suggest to frame the problem a bit more general here.

**Summary:**

The paper considers causal discovery via the post-nonlinear noise model and contributes a novel autoencoder-estimation technique for the multivariate case by applying the same causal order and pruning procedure as proposed for the ANM model.

---

> ### Author Response · Authors · 2021-12-04
> **Response to Reviewer rX9F (1/2)**
>
> Thank you for your review and insightful comments.
>
> We would like to answer your questions and comments in the following. We will reflect the answers in our camera-ready version.
>
> > The authors state that even the publication of the PNL model was already in 2012 there hasn't been work on multi-cause estimators. I found another work ("Post-nonlinear Causal Model with Deep Neural Networks" by Chung etal, 2019) that also addresses a multi-cause PNL with neural networks. Hence, the authors would have to explain the theoretical difference and/or compare them numerically.
>
> Thank you for showing us another existing work. We found the PDF, Y. Chung et al, "Post-nonlinear Causal Model with Deep Neural Networks", class report, 2019. If it is the wrong one, please kindly let us know.
>
> The key difference is that our method aims to estimate causal DAGs among variables while the existing one does not. The existing method estimates PNL models with multiple causes and a single effect by neural networks. However, as the authors discuss in Conclusion, it does not estimate causal DAGs from these results and additional algorithms/methods are required to achieve the DAG estimation.
>
>
> > For the only Theorem the authors state "The “if“ part is obvious from the definition. The converse can be easily verified in the same manner as the ANM case (see Peters et al., 2014, A. 15).". Since there are no space problems here, I would provide the proof for completeness.
>
> We omitted the exact proof of this theorem because it will be long even though the most part is the same as [Peters 14a].
>
> We would show a more detailed sketch in the following.
>
> "If" part:
>
> Let x be a sink variable and X' be the other variables. From the definition of the multivariate PNL model (1), there exists a PNL model such that x=g(f(X')+e), X'⟂e.
> Note that the function f does not do anything for non-parent variables of x.
>
> "Only if" part:
>
> Suppose there exists a PNL model with an effect y that is not a sink on the true DAG G (+). Since y is not a sink, y has children on G. Let z be a sink on the subgraph consisting of y's children and D be descendants of z. Since D does not have children of y, D⟂y | S∪{z} holds. Therefore, from the assumption (+), there exists a PNL model y=g'(f'(S,z)+e'). On the other hand, since z is a child of y, there exists a PNL model z=g(f(S,y)+e).
>
> Given the situation in which S was observed, this contradicts the identifiability of the bivariate PNL model.
>
>
> > There are also no guarantees discussed for how well the autoencoder may fit in a general setting.
>
> In AbPNL, the autoencoder is used to impose the invertibility of the outer function g of the PNL model. Specifically, two neural networks representing g and g^{-1} are arranged in serial and trained so that they become the inverses of the other by minimizing the reconstruction error. This is the general usage of the autoencoder in the field of neural networks. Although we modified AbPNL for our method, the autoencoding part works in the same manner as the original AbPNL.
>
> > The authors mention the issue of stabilization in the Conclusions, but could they elaborate on this also in the Discussion?
>
> Thank you for your suggestion. We will set up the Discussion section and add more detailed discussions.
>
> As stated in the paper, we estimate one PNL model by training the corresponding neural networks several times to stabilize results. Therefore, the stabilization of the training is important for not only improving the performance but also reducing the computational cost. One of the typical approaches is to change the optimizer, while there is a trade-off between stability and speed. Another possible option is to add a regularization term to the loss. To use this idea, we need to consider carefully the impact of changing the loss on using its value when comparing models in the sink identification.
>
> > My main concern is about the numerical comparison. In fact only one single DAG with a few SCMs is considered and the proposed approach compared to the RESIT estimator of ANM and DirectLinGAM. As mentioned above, there seems to be another relevant neural network estimator that could be considered here.
>
> In the experiment in section 4.1, we randomly generated 100 linear and nonlinear problems, respectively, as stated in the caption of  Table 1. In this procedure, not only causal equations but also DAG structures are generated randomly. Two examples in Fig. 2 are (coincidentally) have the same DAG structure.
>
> We apologize for insufficient and misleading expressions. To make it clear, we will add the descriptions in the main text and change the examples in Fig. 2.

---

> > ### Author Response · Authors · 2021-12-04
> > **Response to Reviewer rX9F (2/2)**
> >
> > > The main issue seems to be the extreme computational complexity of the method. Could the authors discuss how this can be alleviated? Also, except for computational complexity no weaknesses are discussed.
> >
> > Thank you for the important point.
> > The computational complexity of the sink identification is mainly due to its iterative manner and model estimations. In a single iteration, we compare k models for given k variables. Therefore, let n be the number of variables, a total of n+(n-1)+...+2=n(n-1)/2-1 model estimations are required, which is the same as RESIT and DirectLiNGAM.
> > Additionally, in our method, we conducted the training of neural networks several times for one model to stabilize the results. Consequently, the total number of training is t*(n(n-1)/2-1), where t is the number of training trials per one model.
> > One effective approach to reduce the computational time is parallelization. Since all the trainings in one iteration can be performed separately, we can reduce the time by parallelizing them, which we used in the experiments. For the time of a single training, the neural networks in our method are used to represent simple scalar functions and do not need to have complex structures. Therefore, the training time is much shorter than the general deep models such as image recognition and language processing. However, as stated in the above response, we think the instability of training is the most important point to be improved.
> >
> > We conducted additional experiments on larger problems of n=15. The following results show the performance of 10 generated problems due to the time limitation. We will increase the number of problems to evaluate the performance in more general and put the results in the camera-ready version with additional analyzes of the results.
> >
> > |               |  d_edit (L) |   N_rev (L) |  d_edit (N) |   N_rev (N) |
> > | ------------- | -----------:| -----------:| -----------:| -----------:|
> > | AbPNL(0.95)   | 11.9 (5.43) | 2.40 (1.71) | 19.2 (4.49) | 3.50 (1.27) |
> > | RESIT(linear) | 2.80 (2.04) | 0.00 (0.00) | 39.2 (12.4) | 5.20 (1.93) |
> > | RESIT(gam)    |  3.3 (1.57) | 0.00 (0.00) | 29.9 (9.17) | 3.90 (2.23) |
> > | RESIT(gp)     | 68.0 (8.01) | 1.60 (1.17) | 32.5 (7.69) | 4.10 (1.52) |
> > | DirectLiNGAM  |  2.5 (1.78) | 0.00 (0.00) | 31.8 (9.54) | 6.70 (2.71) |
> >
> > (L): Results on 10 linear problems, (N): Results on 10 nonlinear problems.
> >
> > > I would also suggest to provide code somewhere.
> >
> > Thank you for the suggestion.
> > We also believe code availability is important and are working on it.
> >
> > > Clarity
> > > The style of writing suffers a bit from English grammar issues.
> >
> > We apologize for the grammar issue.
> > We will carefully check the paper.
> >
> > > Further comments
> > > "Causal discovery methods assume that causal structures follow structural equation models" What about constraint/score-based methods. These do not assume one particular SEM, rather a very general class. I suggest to frame the problem a bit more general here.
> >
> > Thank you for the suggestion.
> > We will revise the abstract to state the problem in more general.

---

### Comment · Area_Chair_p1Vj · 2021-12-20
**Disagreement among reviewers**

There are some significant differences among the reviewers, both in the scores, and whether or not it is above threshold for publication. Reviewer wyQs, who voted that it was below threshold for publication, had three cons for the paper: (i) lack of novelty, (ii) no proof of identifiability or consistency, and (iii) the experiments are too simple. The authors claim that the identifiability and consistency results follow from previous results in the literature, and (ii) they have done further experiments on larger numbers of variables. Does their reply resolve your worries about these issues?

With respect to the question of novelty, the question would seem to be whether putting together some slightly modified existing pieces of research in such a way as to produce the only algorithm that can be applied to a general post-nonlinear model is sufficient.

---

### Decision · Program_Chairs · 2022-01-12

**Decision:**

Accept (Poster)

**Comment:**

A Multivariate Causal Discovery based on Post-Nonlinear Model This paper modifies an algorithm for searching for additive noise models to searching for multivariate post-nonlinear models. The paper also tests the algorithm on synthetic (100 random DAGs and equations) and real data. The authors have agreed to make the code public. They use an autoencoder based estimator of Uemura and Shimizu to the multi-cause case. Simulations show the superiority of the new multivariate PNL algorithm over algorithms that make more restrictive assumptions in the non-linear case (although it performs worse than DirectLiNGAM in the linear case). THe algorithm was also tested on a well-known carefully constructed simulation of fMRI data. In response to the reviewers they have added simulations on larger numbers of variables (15) and analyzed the complexity of the algorithm.

The main argument against the paper is that it is a fairly simple adaption of already existing techniques, so its theoretical novely is quite low. On the other hand, its practical novelty is much higher, since it is the only described and implemented algorithm for searching for multi-variate PNL causal models.

Two of the reviewers voted a 6 and one voted an 8. I concur that it should be accepted, given the practical usefulness of the algorithm and the tests on the simulated and real data.